# Defining and Quantifying National-Level Targets, Indicators and Benchmarks for Management of Natural Resources to Achieve the Sustainable Development Goals

**Chris Dickens** [1,*] , **Vladimir Smakhtin** [2], **Matthew McCartney** [3], **Gordon O'Brien** [4] **and Lula Dahir** [1]

[1] International Water Management Institute, Pretoria 0184, South Africa; lula.dahir@gmail.com
[2] United Nations University—Institute for Water, Environment and Health (UNU-INWEH), Hamilton, ON L8P 0A1, Canada; vladimir.smakhtin@unu.ed
[3] International Water Management Institute, IWMI-SEA, Vientiane PO Box 4199, Laos; m.mccartney@cgiar.org
[4] School of Biology and Environmental Sciences, Faculty of Agriculture and Natural Sciences, University of Mpumalanga, Private Bag X11283 Nelspruit, South Africa; gordon.obrien@ump.ac.za
* Correspondence: c.dickens@cgiar.org; Tel.: +27-83-269-6207

**Abstract:** The 2030 Agenda for Sustainable Development, the Sustainable Development Goals (SDGs), are high on the agenda for most countries of the world. In its publication of the SDGs, the UN has provided the goals and target descriptions that, if implemented at a country level, would lead towards a sustainable future. The IAEG (InterAgency Expert Group of the SDGs) was tasked with disseminating indicators and methods to countries that can be used to gather data describing the global progress towards sustainability. However, 2030 Agenda leaves it to countries to adopt the targets with each government setting its own national targets guided by the global level of ambition but taking into account national circumstances. At present, guidance on how to go about this is scant but it is clear that the responsibility is with countries to implement and that it is actions at a country level that will determine the success of the SDGs. Reporting on SDGs by country takes on two forms: (i) global reporting using prescribed indicator methods and data; (ii) National Voluntary Reviews where a country reports on its own progress in more detail but is also able to present data that are more appropriate for the country. For the latter, countries need to be able to adapt the global indicators to fit national priorities and context, thus the global description of an indicator could be reduced to describe only what is relevant to the country. Countries may also, for the National Voluntary Review, use indicators that are unique to the country but nevertheless contribute to measurement of progress towards the global SDG target. Importantly, for those indicators that relate to the security of natural resources security (e.g., water) indicators, there are no prescribed numerical targets/standards or benchmarks. Rather countries will need to set their own benchmarks or standards against which performance can be evaluated. This paper presents a procedure that would enable a country to describe national targets with associated benchmarks that are appropriate for the country. The procedure builds on precedent set in other countries but in particular on a procedure developed for the setting of Resource Quality Objectives in South Africa. The procedure focusses on those SDG targets that are natural resource-security focused, for example, extent of water-related ecosystems (6.6), desertification (15.3) and so forth, because the selection of indicator methods and benchmarks is based on the location of natural resources, their use and present state and how they fit into national strategies.

**Keywords:** water resources; natural resources; resource security; SDGs; goal; target; benchmark; standard

## 1. Introduction

In 2015 the UN General Assembly and Heads of States adopted the 2030 Agenda for Sustainable Development, a global development agenda that lays out 17 Sustainable Development Goals (SDGs) to be achieved by 2030. The 2030 Agenda calls for a long-term transformation that balances the three dimensions of sustainable development-social, economic and environment in a holistic and coherent manner.

The 2030 Agenda has 17 SDGs underpinned by 169 targets that are described by >230 indicators with associated methods. However, this overall system is aspirational and most of the targets do not have numerical values for the globe. The 2030 Agenda [1] states that it is the responsibility of countries to contextualise the Goals and Targets for their own purpose and that it is implementation at a country level that will be the key to success of the SDGs. Below are the relevant statements from the 2030 Agenda document:

"Targets are defined as aspirational and global, with each government setting its own national targets guided by the global level of ambition but taking into account national circumstances. (55). We recognise that there are different approaches, visions, models and tools available to each country, in accordance with its national circumstances and priorities (59). The Goals and targets will be followed-up and reviewed using a set of global indicators. These will be complemented by indicators at the regional and national levels which will be developed by member states (75)". [1]

Note on terminology: there is general confusion around the definitions of terms describing goals, objectives, targets, indicators, benchmarks and so forth the hierarchy of use of selected terms in this paper is shown in Table 1 with greater detail and definitions given in the Appendix A. These terms, where possible, align with the 2030 Agenda.

**Table 1.** Hierarchy of terms used in this paper, designed to align with 2030 Agenda. The hierarchy starts with the most over-arching term (vision) and ends with the most directed and specific (benchmark). Greater detail and definitions are provided in Appendix A (SDG–Sustainable Development Goal).

| | |
|---|---|
| 1 | Vision for the resource |
| 2 | Goal (including SDG) |
| 3 | Target |
| 4 | Indicator and indicator method |
| 5 | Benchmark (numerical value) |

The UN Statistical Commission provides a global indicator framework as a mechanism for implementing the 2030 Agenda. While the Goals and targets are described in the 2030 Agenda document itself [1], the detail of the indicators has been managed by the UN Statistical Commission through the IAEG (Inter-Agency Expert Group on the SDGs) which publishes updates on the accepted methods at regular intervals [2]. The step-by-step indicator methods for implementation as provided by the IAEG are thus the norm for global reporting and have been disseminated for use at country level (e.g., Goal 6 (water) and all its indicators are available on the UN Water web site http://www.sdg6monitoring.org/indicators/). However, due to the unique and uneven distribution of natural resources as well as stresses on these natural resources within each country, the aggregated global index data is of limited value for in-country management. Thus the 2030 Agenda makes provision for

Voluntary National Reviews (VNR) where countries can report to the UN using their own unique data. As shown above, the 2030 Agenda encourages countries to set their own unique national level goals, targets, indicators and benchmarks for their own internal management processes, while these may be aligned with the spirit of the 2030 Agenda goals and targets. Thus, each country will need to produce two different types of SDG data, the first being for global reporting where the indicator methods are prescribed and the second being country-unique data which can be used for their VNR and also for management of resources.

Both the global and national targets and indicators are generally narrative and need to have quantifiable measures (benchmarks) put in place that can be used to assess performance towards sustainable development. Thus, to take an example, SDG Target 6.6 states "By 2020, protect and restore water-related ecosystems, including mountains, forests, wetlands, rivers, aquifers and lakes" but at a global level there is no provision for any quantitative measure for achievement. While this target provides a general direction for every country, it is up to each country to set clear management targets and benchmarks so, for example, one country may decide to protect 100% of its wetlands as its commitment to Target 6.6, while another may choose to protect only 20%, based on its own particular context. In both cases, each country should justify why this benchmark has been chosen and how this aligns with the 2030 Agenda and ultimately with sustainable development. These benchmarks should be aspirational yet attainable; context specific yet adaptable to change and finally, based on evidence [3].

A country may then decide that in addition to monitoring the global 6.6.1 indicator, it also needs to have unique indicators that, given its specific national context, are more appropriate for monitoring the state of its own unique ecosystems. Reporting of these country unique results will then form part of the country's VNR. According to [4] VNRs may examine the agreed global indicators for SDGs and related targets but countries may also choose to refer to complementary national and regional indicators. It is these additional country level indicators and benchmarks that are the purpose of this report.

### 1.1. Defining Targets and Benchmarks to be Sustainable

The mission of the 2030 Agenda is sustainable development. It is thus important that the targets and benchmarks accord with sustainability principles, that is, that by achievement of these targets, this indicates that the resource is indeed being managed in a sustainable way. To do this, it is necessary to understand the concept of sustainability.

Leopold in 1949 [5] stated that sustainable development is the organising principle for sustaining finite resources necessary to provide for the needs of future generations of life on the planet. It is a process that envisions a desirable future state for human societies in which living conditions and resource use continue to meet human needs without undermining the "*integrity, stability and beauty*" of natural biotic systems. This early definition gives emphasis to an overriding principle of sustainable development, the need to balance the use and protection of resources.

The Brundtland Commission [6] defined sustainable development as "development that meets the needs of the present without compromising the ability of future generations to meet their own needs." This definition served as a basis during the United Nation's Earth's Summit meeting (Rio, UN, 1992), the World Summit on Sustainable Development [7] and the UN Conference on Sustainable Development [8]. The Brundtland definition of sustainable development comprises two key elements: needs—aiming at meeting the needs of all humans in the present and future and limits—limitations to environmental resources and the biosphere's ability to meet needs of present and future generations. Sustainable development was divided during the World Summit in Johannesburg into the three pillars of environment, economy and social [7], the intention being that sustainability can be attained by balancing these three dimensions. The environmental pillar is made up of the land, water and air resources, the physical, chemical and biological components which interact to form functional ecosystems.

The report "The Future We Want" from which the SDG programme evolved, defined sustainable development as:

> "promoting sustained, inclusive and equitable economic growth, creating greater opportunities for all, reducing inequalities, raising basic standards of living; fostering equitable social development and inclusion; and promoting integrated and sustainable management of natural resources and ecosystems that supports inter alia economic, social and human development while facilitating ecosystem conservation, regeneration and restoration and resilience in the face of new and emerging challenges".
>
> The Future We Want [8]

It could be argued that this definition gives greater emphasis to development and less to securing natural resources when compared to those of Leopold [5] and the Brundtland Report [6]. But indeed, it is the trade-off between the need to develop and the need to protect natural resources or natural capital that is the enduring dilemma in sustainable development, made more complex by spatial scales as resources and developments are globally heterogeneous.

The first global report on SDG indicators for water [9] has reported that we are "not on track" to meet the SDG targets. Those targets and indicators that focus on natural resources (or natural capital) are key SDGs, as without sustainable use of these natural resources, all of the other SDGs would be compromised [10]. Wackernagel et al [10] have raised the caution that the 2030 Agenda is already weak in balancing between natural resources and development, as it places more emphasis on indicators related to social development, while those related to resource security receive substantially less attention (only 13.6% of the indicators). This they argue will hamper the success of the SDGs because the advancement of human wellbeing depends on benefits derived from natural resources. The balance between development and ecosystem resources is documented by Wackernagel et al [10] in their National Footprint Accounts approach that compares an Ecological Footprint to the UN Human Development Index, the results showing that few countries have yet achieved a balance that could be classified as sustainable. Further concern was raised by UN-DESA [11] who noted for SDG 15 that:

> "Five indicators of response all show positive trends, that is, that efforts towards implementation of SDG 15 are increasing. On the other hand, SDG indicators 15.1.1 and 15.5.1, the two indicators on the state of life on land, both show declines". They state: "Understanding why the overall state of nature is declining despite increasing efforts towards conservation and sustainable development is an urgent priority if SDG 15 is to be met".
>
> UN-DESA [11]

What is clear from these perspectives, is that targets and indicators, at a global as well as at a country level, need to consider both natural resources and the society that benefits from them.

How does society inform the decisions that need to be made, between the obvious need of a burgeoning society to prosper socially and economically, with the limits that are imposed by the availability of natural capital? Contributing a key perspective when evaluating trade-offs is the Planetary Boundaries approach developed by Rockström et al [12] which has pointed to thresholds in the Earth's system beyond which irreversible changes might have enormous impacts on humanity's survival. These thresholds "exist irrespective of people's preferences, values or compromises based on political and socioeconomic feasibility." Raworth [13] developed the planetary boundary concept further by including societal needs while still considering the safe operating boundaries of the ecosystem. To define what is acceptable to both society and the ecosystem, Ostrom [14] advised that relationships between different social-ecological systems should be identified and analysed, highlighting the trade-offs between the ecosystem and the needs of society. As global populations increase and the stress on the natural resources is intensified, trade-offs have become a critical issue for sustainable development but these trade-offs need to be informed by the likes of the Planetary Boundaries which may provide the limits within which development may take place.

*1.2. A Procedure for Setting Targets and Benchmarks*

The process of defining national-level targets and setting national-level benchmarks, has not been fully articulated, nor well understood as existing guidelines tend to focus on the higher policy level. National governments must decide how to incorporate SDG targets into national planning processes, policies and strategies and set their own targets, taking into account local circumstances [1]. This paper suggests "a simplified procedure for countries to develop contextually relevant national targets that can be aligned with global SDG targets and are supported by contextually relevant national indicator methods with associated benchmarks". The procedure may be used by countries to set national targets that are not only useful for local management purposes but also for global SDG reporting. It is aimed primarily at those SDG targets and indicators that have a basis in protection and sustainable use of natural resources and which require that the state of natural resources be monitored (such indicators make up only 13.6% of the SDG indicators, [10]). Those SDG targets that are most suited to implementation of this procedure are those that seek to protect land and water resources (see below) while other indicators may also benefit from the essence of this approach:

- 2.4 . . . sustainable food production systems . . . help maintain ecosystems . . . progressively improve land and soil quality.
- 6.3 . . . improve water quality . . . (in particular 6.3.2 on ambient water quality that targets "good" water quality).
- 6.6 . . . protect and restore water-related ecosystems, including mountains, forests, wetlands, rivers, aquifers and lakes.
- 12.2 . . . achieve sustainable management and efficient use of natural resources.
- 14.1 . . . prevent and significantly reduce marine pollution of all kinds . . . .
- 14.2 . . . sustainably manage and protect marine and coastal ecosystems . . . to achieve healthy and productive oceans.
- 14.3 . . . minimize . . . ocean acidification . . .
- 14.4 . . . restore fish stocks . . . produce maximum sustainable yield as determined by their biological characteristics.
- 15.1 . . . conservation, restoration and sustainable use of terrestrial and inland freshwater ecosystems and their services...
- 15.2 . . . sustainable management of all types of forests, halt deforestation, restore degraded forests and substantially increase afforestation and reforestation globally.
- 15.3 . . . combat desertification, restore degraded land and soil . . . desertification, drought and floods . . . strive to achieve a land degradation-neutral world.

## 2. Materials and Methods

The procedure below (Figure 1) is designed to assist national governments to quantitatively describe natural resources in order that benchmarks can be set as objectives for management of natural resources. The procedure sets out to define the resource to be managed (Step 1); to describe the vision that society has for that resource (Step 2); to prioritise the areas requiring monitoring (Step 3); to prioritise the indicators that best describe the resource (Step 4); to quantify potential indicator based benchmarks (Step 5) and to refine these together with stakeholders prior to adoption (Steps 6 & 7).

The procedure has its foundation in the procedures followed in Australia and New Zealand [15], the logical framework process of setting targets used by United Nations Economic Commission for Europe and the World Health Organization [16] and the South African procedure for determination of Resource Quality Objectives [17].The latter publication provides decision support tools that assist with the prioritisation of geographic areas for monitoring (the Resource Unit Prioritisation Tool) and the selection of indicators and measures for target development (the Resource Unit Evaluation Tool).

The procedural steps below have been designed to guide the process of establishing a balance between the need to protect and use resources at multiple spatial scales ranging from sites, management

areas, regional to national scales. They thus consider both the socio-economic context of the geographic areas as well as their ecosystem/resource characteristics and use this context to describe locally relevant targets, indicators and numerical benchmarks that can be used to monitor the targets. The procedure advocates an adaptive management approach so that the benchmarks produced are seen as hypotheses, subject to change if it is found that management has not achieved the desired balance between the use and protection of the resources or if the desired vision for the resource is not being achieved. This procedure is suitable for any country wishing to set target and benchmarks for the management of natural resources and is of particular value in aligning national targets with Agenda 2030 on Sustainable Development.

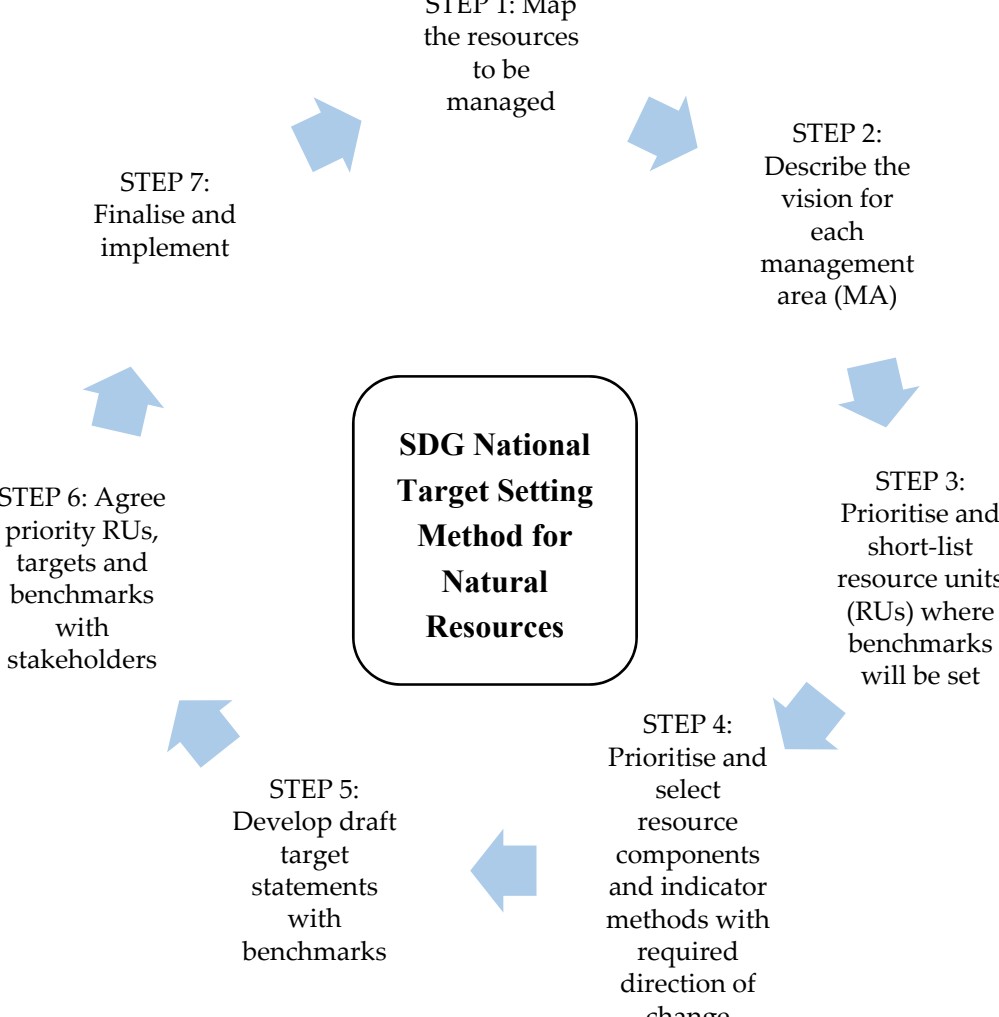

**Figure 1.** Schematic of the National SDG Target Setting Method for Natural Resources. A full description is given below.

Step 1. Describe the resources to be managed within a spatial dimension

1.1 Divide the country or region into Management Areas (MAs), that is, relatively homogenous geographic areas where management objectives may be uniformly applied (e.g., a river basin or administrative area). Map available information describing the socio-ecological situation, for example, land-cover, ecosystems, spatial distribution of the resource in question (e.g., water, forests etc), population, development, social and economic issues, land-use and so forth

1.2 Select the associated SDG Goal and targets (e.g., Goal 6: targets 6.3, 6.4 and 6.6) that need to be locally managed in order to support sustainable development.

1.3 Sub-divide each MA into Resource Units (RUs), that is, geographical areas containing natural resources (e.g., different reaches of a river) that are sufficiently distinct to warrant their own specification of requirements. A RU is the basic unit used for management of a natural resource to which targets will apply. There will be multiple RUs for each MA and each RU will be specific to its resource (i.e., a freshwater RU may be different to a forest RU).

Step 2. Describe the vision for each Management Area

The vision or management objectives for each MA should consider both the need to use and to protect the resources. While stakeholder involvement in establishing a vision is important, such a process may already be reflected within policy documents. Where a fresh visioning process conflicts with policy, then this would need to be resolved at a political level.

2.1 Extract the vision or management objectives for each resource from national policy, strategic or management plans and so forth that are relevant for each MA. Alternately follow a separate stakeholder engagement procedure for setting a vision or management objective for the resource [18,19].

2.3 Associate the vision for each resource with the relevant SDG targets (e.g., the vision for ambient water quality in a MA will be associated with SDG 6.3.2)

2.4 For each transboundary natural resource, harmonise the vision with neighbouring countries as necessary.

Step 3. Prioritise and short-list RUs where benchmarks will be set

In the interest of cost-effectiveness, it is only necessary to monitor and set targets for key RUs within a MA, that is, those RUs that have the most important ecosystems, which are subject to the greatest stress and may be the most impacted. The procedure to determine this follows:

3.1 From Step 1.1. extract and map socio-economic and ecological/resource information for each RU.

3.2 Assess the importance of the ecosystem services and resources within each RU to users and to the wider ecosystem.

3.3 Describe the activities in the RU that pose a threat to the continued supply of these ecosystem services and resources to users (e.g., irrigation will remove water from a river).

3.4 Identify the components of the ecosystem that will be impacted by these activities (e.g., river flow and all associated ecosystem characteristics).

3.5 Rank the level of threat to these ecosystem components within each RU.

3.6 Consider practical considerations associated with setting targets within each RU (e.g., access for sample collection, location of existing data points etc).

3.7 Based on the relative ranking and weighting of each of the above criteria, select priority RUs using prioritisation scores. Targets will only be set for the priority RUs. There is no "correct" number of RUs that should be selected for monitoring as selection may be tempered by available budget but it is also possible to conduct statistical tests to confirm when adequate coverage is attained.

Step 4. Prioritise and select resource components and indicator methods to be used for monitoring and propose the direction of change to improve sustainability

In the interest of cost effectiveness, while in keeping with the vision for the resource that has been set, it is prudent to monitor only the important components of the ecosystem/resource within each priority RU and thus a process needs to be followed to prioritise the components and associated indicators. The procedure to do this follows:

4.1 Gather data to assess the present state of the resource. Follow this by interpreting the existing level of impact and likely changes that have or will occur due to current and planned future use on those components of the resource identified in Step 3.4 above (e.g., changes to water volumes, water quality, biota and so forth).

4.2 Identify the important users of the natural resources in the RU and document their requirements in relation to each of the resource components in Step 4.1. Alongside this and in keeping with the vision for the area, document the need to protect these same natural resources.

4.3 Rank the resource-components with associated indicator methods that best describe the management target (e.g., in a particular RU it may be that nutrient accumulation in the river is the most important component, in which case soluble phosphorus and total inorganic nitrogen may be the most appropriate indicators for monitoring). Select suitable indicator methods for measurement of each of these components (e.g., a method for nutrient monitoring).

4.4 Establish the desired direction of change from the present condition, for each of the selected resource-components in order to promote sustainability. Thus, for some components the plan would be to maintain the status quo, while for others a strong improvement would be necessary. In situations where the objectives for the system are to expand the use, then the direction of change may negative.

Step 5. Develop draft target statements with benchmarks

For each of the resource-components with the appropriate indicator method as prioritised in Step 4, it is necessary to define a target narrative and also benchmarks that need to be achieved in order for the vision or management objectives for the RU (as defined in Step 2) to be realised. SDG natural resource targets are all narrative and similarly national level targets should also be narrative (Step 5.1). These then need to be supported by quantifiable benchmarks (Step 5.3).

5.1 Draft narrative target statements for each priority component (from Step 4) for each RU (from Step 3), that fit with the vision and management objectives (from Step 2), for example, in response to a vision to maintain healthy water in a lake for tourism purposes, the target narrative may state that "nutrient concentrations in this lake need to be at concentrations low enough to maintain clear water and limit blooms of algae."

5.2 Numerically quantify the present state of the priority resource components determined in Step 5.1, for example, the annual average phosphorus concentration has been 0.06 mg/L.

5.3 Based on the vision for the resource, the target narrative, the necessary direction of change and the best scientific information available, set appropriate draft benchmarks for each priority resource-component. The benchmark provides an objective measure that supports appropriate management, for example, the benchmark average annual phosphorus concentration in the lake should be less than 0.03 mg/L.

Step 6. Agree priority Resource Units, targets and benchmarks with stakeholders

Steps 3–5 have been largely scientific, so it is appropriate to present and consider the results so far with stakeholders from each MA with allowance for improvement. Stakeholder involvement is important in the process of setting natural resource goals and targets as it promotes their support.

6.1. Present and refine with stakeholders (each country would determine which stakeholders are appropriate), the RU prioritisation, the priority resource-components and indicators, the present state, the proposed direction of change, the target narratives and the benchmarks for achievement of the target narrative.

Step 7. Finalise and implement

7.1 Resource managers collate the final results into management plans, policies and even into law and implement as appropriate to the country. Note that performance should be evaluated in terms of progress towards the goals, targets and benchmarks, while the benchmarks should not be considered as pass/fail standards.

## 3. Results

While no purposive example yet exists for the direct implementation of this procedure, there is precedent directly from implementation of the Resource Quality Objectives procedure in South Africa,

the model that provided the greatest input to the procedure in this paper [17]. The procedure has been implemented in important sub-basins on the Vaal River [20] and the Olifants River [21] in South Africa, while it has been less closely followed in a number of other basins.

1.  Key success factors of this approach include i) The vision for the resource sets the context within which RQOs (targets) and benchmarks can be set; ii) Geographical areas (Resource Units) are prioritised for monitoring, thus those areas where there is greatest need for close attention by management (where there are high demands or where the ecosystem is fragile, for example) are given priority and will have targets and benchmarks to provide management with direction; iii) indicators are prioritised to ensure that both the protection and the use aspects of the resource are included in the targets; iv) Based on the present-day status of each of these indicators, the required trajectory of change in order to meet the vision is determined. From this and making use of literature, guidelines for water quality, models and so forth, a value for each target (benchmark) is recommended and ultimately is written into law ([22]). These target values are aspirational but realistically attainable. Success in performance of management is when the quality of the resource has exceeded the target benchmark or where progress is in a positive direction towards doing so.

Table 2 presents an example of targets (Resource Quality Objectives) and benchmarks (Numerical Limits) from implementation of the South African [17] procedure in the Olifants River basin in South Africa, the table is extracted adapted from the recommendations [21] that were amended before documenting in the government gazette that enters these benchmarks into law [22]. This government gazette contains targets for relevant aspects of the water resource, including the discharge (flow) of water in the river during both the dry and wet seasons, the quality of the water, in particular the nutrients, salts, toxins and some physical variables (sediment etc). It also sets targets and benchmarks for instream and riparian habitats. There are also benchmarks for impoundments, for water levels, habitats and water quality, as well as for fish. Benchmarks are also set for groundwater aquifers, requiring simply that there should not be a negative trend to groundwater levels. The components included in this government gazette did not fully represent the proposed targets and benchmarks that had included river biota as well as several indicators for vegetated wetlands, as the implementing agency had to limit the scope of the monitoring programme that would follow.

## 4. Discussion

The SDG goals and targets now come to centre-stage in the sustainable development narrative. The global society is increasingly going to resort to these goals and targets for evidence on progress in relation to sustainable development and for this, spatial scale and quantifiable monitoring is imperative. Within the context of what the resource was in its natural state and what the state is at the present time, then it becomes essential for the world and for countries to set objective targets against which progress can be monitored towards a future state that is indeed sustainable and better still, is aspirational and will provide more benefits to society than borderline sustainability. These are the decisions that stakeholders need to make, that is, just how many benefits need to be available from natural resources but within the limits set by the Planetary Boundary concept [12].

It should be noted that while the Planetary Boundaries [12] describe the limits of acceptable use of natural resources, these are not the same as the benchmarks required to achieve SDGs. The thresholds described by Rockström et al. are between sustainability and unsustainability, while the SDG benchmarks, especially when applied at a country level, are aspirational and thus should be substantially "better" than the safe operating boundaries of the ecosystem as defined [12].

**Table 2.** Example of targets (equivalent to RQOs) and benchmarks (numerical limits) determined for the Olifants River in South Africa (equivalent to a MA), adapted from the Department of Water and Sanitation South Africa [21].

| RU | REC | Component | Sub Component | Target Narrative (Equivalent to RQOs or Resource Quality Objectives) | Indicator | Benchmark (Numerical Limits) | | |
|---|---|---|---|---|---|---|---|---|
| | | | | | | | Maintenance low flows ($m^3/s$) (%ile) | Drought flows ($m^3/s$) (%ile) |
| Klein Olifants (EWR site 3) | C | Water quantity | Low flow | Low flows should be improved in order to maintain ecosystem functioning and ecotourism. | Environmental FlowNatural MAR = $81.54 \times 106$ $m^3$ | Oct | 0.135 (70) | 0.071 (99) |
| | | | | | | Nov | 0.227 (80) | 0.100 (99) |
| | | | | | | Dec | 0.313 (80) | 0.160 (99) |
| | | | | | | Jan | 0.394 (80) | 0.200 (99) |
| | | | | | | Feb | 0.467 (80) | 0.237 (99) |
| | | | | | | Mar | 0.384 (80) | 0.161 (99) |
| | | | | | | Apr | 0.324 (70) | 0.162 (99) |
| | | | | | | May | 0.257 (70) | 0.119 (99) |
| | | | | | | Jun | 0.200 (70) | 0.103 (99) |
| | | | | | | Jul | 0.167 (70) | 0.087 (99) |
| | | | | | | Aug | 0.134 (70) | 0.070 (99) |
| | | | | | | Sep | 0.112 (70) | 0.046 (99) |
| Outlet of quaternary-outlet of IUA9 | D | Water quality | Nutrients | Nutrients need to be minimized in order to ensure that the system is maintained in a mesotrophic condition. | Nitrate ($NO_3$) | $\leq 4.00$ mg/L N | | |
| | | | | | Phosphate ($PO_4$) | $\leq 0.125$ mg/L P | | |
| Outlet of quaternary-outlet of IUA8 | B | Biota | Fish | Fish communities should be improved to a good condition and should include viable populations of ecologically important species | State of fish population according to the FRAI (Fish Response Assessment Index) | FRAI score B | | |

The vision for the river is described in terms of the Recommended Ecological Category (REC) whereby an "A" category ecosystem is in its natural state, and a "D" category ecosystem is "largely modified" and on the brink of collapse (following the procedure of [23]).

In order to select indicators and benchmarks for the indicators that will adequately show progress in management of natural resources, there is already some precedent that supports the procedure that has been recommended in this paper. These country examples provide useful guidance to a country implementing the approach presented in this paper, as their examples provide detail of many of the recommended steps. Most countries do not however have such procedures and thus may find the procedure in this paper useful. Where a country does have a procedure, this paper may be of value in reviewing their approach to confirm that it addresses the kind of objectives management approach that is necessary for Agenda 2030 to be a success.

The South African procedure [17] that provided the foundation for the steps presented in this paper has been implemented in a number of basins [21] but most importantly in the Olifants and Vaal Rivers, two high-profile water resource management areas in South Africa. In both of these basins there is conflict between the use and protection of resources, with society fervently trying to develop and, in the process, using and abusing natural resources. In both of these basins, clear targets and benchmarks (Resource Quality Objectives and Numerical Limits in the local context) were successfully developed including strong stakeholder participation and in 2015 these were legally gazetted in a Government Notice [22]. Implementation of these objectives is however problematic, as for example, in 2018 a constitutionally mandated Human Rights Commission reported a "prima facie violation of the right of access to clean water, a clean environment and human dignity [24], resulting from continuing and rampant pollution of the Vaal River.

The Australian and New Zealand Guidelines [15] provide an authoritative guide for setting water quality objectives required to sustain current or likely future environmental values for natural and semi-natural water resources in Australia and New Zealand. This approach allows flexibility at a local level to set objectives (benchmarks) appropriate for the region. These objectives are the recommended limits to acceptable change in water quality that will continue to protect the associated environmental values. However, they are not mandatory and have no formal legal status. The guideline states that water quality objectives are the specific water quality targets agreed between stakeholders or set by local jurisdictions, that become the indicators of management performance. Normally, only those indicators considered relevant to the environmental issues or problems facing the resource are selected for deriving water quality objectives. They serve to protect the designated environmental values of a resource. The Australian and New Zealand Guidelines [15] recommend a number of steps to reach clear objectives (targets with benchmarks), that is, (i) identify the environmental values that are to be protected in a particular water body and the spatial designation of the environmental values; (ii) identify management goals and then select the relevant water quality guidelines for measuring performance. Based on these guidelines, set water quality objectives that must be met to maintain the environmental values; (iii) develop statistical performance criteria to evaluate the results of the monitoring programs (e.g., statistical decision criteria for determining whether the water quality objectives have been exceeded or not), (iv) develop tactical monitoring programs focusing on the water quality objectives; (v) initiate appropriate management responses to attain (or maintain if already achieved) the water quality objectives/targets. Countries can make use of frameworks such as the Set Ecological Targets (SET) framework used in Australia [25] as a tool to support multi-stakeholders to define environmental and human values to minimize environmental impact and subsequently set targets.

The EU Water Framework Directive has also provided important precedent. For example, the UK Technical Advisory Group [26] provides the UK interpretation in terms of setting objectives for rivers, lakes, groundwater, estuaries and coastal waters. Objectives/targets are set for water bodies and expressed in terms of status. Thus, the objectives may include narrative statement such as; "preventing deterioration in status; restoring water bodies to good status by *x* date"; standards (benchmarks) are then matched to the objectives of the Directive. The environment agencies use standards to set limits on the amount of water that can be abstracted or how much pollutant can enter the environment, to either improve environmental quality or to prevent it from deteriorating. The Directive requires

that the overall status of the water body is determined by the lowest status from all the standards (indicators) that are assessed. This is known as the 'one out, all out' rule. To have high status, for example, a water body cannot fail any of the standards (indicators) associated with high status. The Directive provides a step procedure for the determination of standards for specific pollutants.

Recently UN Environment [27] published a guideline that gives a generic protocol for setting targets but this does not provide any detail. The US Environmental Protection Agency has developed guidelines for deriving numerical national water quality criteria {28} for the protection of aquatic organisms and their uses and they provide the method for doing this and recommend that states may use these guidelines to derive water quality standards (benchmarks). The National Recommended Water Quality Criteria [28] table presents a table of the "highest concentration of specific pollutants or parameters in water that are not expected to pose a significant risk to the majority of species in a given environment." However, production of such standards is divorced from the local context. It remains for the State to determine the most acceptable benchmark to ensure sustainability of the water resource, a task that can be undertaken using the procedure recommended in this paper.

## 5. Conclusions

A procedure has been recommended in this paper that supports countries to develop contextually relevant national targets that can be aligned with and inform global SDG targets and are supported by contextually relevant national indicator methods with associated benchmarks. Principles from *The Future We Want* [8] and *Transforming our world: the 2030 Agenda for Sustainable Development* [1] provide the philosophical foundation for the procedure while procedures and implementation guidelines from several other agencies are considered to ensure global consistency and good practice.

The final call of the 2030 Agenda states that:

"We encourage all member states to develop as soon as practicable ambitious national responses to the overall implementation of this Agenda. These can support the transition to the SDGs and build on existing planning instruments, such as national development and sustainable development strategies, as appropriate".

([1], item 78)

The call is thus out for this adaptation to go forwards but for the resource orientated SDGs, there is no guidance or procedure available to do this. This paper provides such guidance. The procedure in this paper has been designed in such a way that it can be used to define the targets, prioritise the resources to be monitored, select indicator methods and set benchmarks for any resource. The procedure remains constant but the country-level prioritisation of resource-components to align with the global target, the indicator methods and benchmarks will vary for different resources.

Agenda 2030 states that the SDGs are aspirational ([1] Item 55), thus monitoring and compliance entails a systematic process to measure and manage performance in management of the resource towards achievement of the vision, goal, target and benchmarks. Compliance with benchmarks would be achieved when the resource is equal to or in a "better" condition than indicated by the benchmark or when there is evidence that the resource quality (as indicated by indicator values) is moving towards the target and not away from it. In the event that there is a change in direction away from the target, then it indicates that the measures in place to manage the resource are not sufficient to bring the resource into alignment with the target or alternately that the target was not reasonable in which case a new target needs to be set following the full process described (including stakeholder consultation etc). Care must be taken however, to ensure that there is not just a systematic lowering of the target expectations if compliance is not achieved. The SDG targets should remain aspirational and should be clear in their orientation towards sustainability.

**Author Contributions:** C.D. conceptualized the paper and drafted most of the text. V.S., M.M. and G.O. provided critical review, built on the conceptualization and provided text. L.D. drafted the initial versions and accessed the necessary literature.

**Funding:** This research was funded by the Water, Land and Ecosystems (WLE) program that forms part of the CGIAR (see acknowledgements), as part of the VCR Flagship Project (VCR—*Managing resource variability, risks and competing uses for increased resilience*).

**Acknowledgments:** The International Water Management Institute (IWMI) and the CGIAR Research Program on Water, Land and Ecosystems (WLE). The CGIAR Research Program on Water, Land and Ecosystems (WLE) combines the resources of 11 CGIAR centres, the Food and Agriculture Organization of the United Nations (FAO), the RUAF Foundation and numerous national, regional and international partners to provide an integrated approach to natural resource management research. WLE promotes a new approach to sustainable intensification in which a healthy functioning ecosystem is seen as a prerequisite to agricultural development, resilience of food systems and human well-being. This program is led by the International Water Management Institute (IWMI) and is supported by CGIAR, a global research partnership for a food-secure future).

**Conflicts of Interest:** The authors declare no conflict of interest. The funders had no role in the design of the study; in the collection, analyses or interpretation of data; in the writing of the manuscript or in the decision to publish the results.

## Appendix A

There has been protracted use of the terms vision, goal, objective, target etc in the literature. The use in relation to development matters has become confusing and terms are often used interchangeably and indeed the advent of the 2030 Agenda [1] has added further to that confusion by failing to define these terms. The 2030 Agenda has not continued the use of the terms objective, threshold or benchmark. While each of these may have its merits, this paper builds on the 2030 Agenda and suggests a hierarchy and definition of these terms (Table A1).

**Table A1.** Proposed hierarchy of terms used in this paper, designed to align with the 2030 Agenda.

| Term | Use at a Global (GL) and National (NL) Level |
| --- | --- |
| Vision—"An aspirational description of what an organization would like to achieve or accomplish in the mid-term or long-term future. It is intended to serve as a clear guide for choosing current and future courses of action" [29] | GL, for example, 2030 Agenda "We envisage a world free of poverty, hunger, disease and want..." NL, for example, water in a river basin is used productively but sustainably for agriculture and tourism. |
| Goal—"the desired result of management in accordance with the aspirations of the Vision. | GL, for example, SDG Goal 6 Ensure availability and sustainable management of water and sanitation for all. NL—Countries should adopt the global goals. |
| Target—"something that you are planning to do or achieve" [30]. | Thus, a target is a statement of something planned in order to reach the Goal. Objective is a common synonym. e.g., SDG 6.6 "By 2020, protect and restore water-related ecosystems, including mountains, forests, wetlands, rivers, aquifers and lakes." NL, for example, by 2020, protect and restore water resources provided by the Inner Niger Delta (in Mali) in particular the annual flood and distribution of natural vegetation to ensure that local people continue to benefit. |
| Indicator—meaningful, simple and quantifiable metric or method used to measure progress towards the target | The UN has provided >230 indicator methods that need to be used to measure progress to the SDG targets. GL, for example, Global map of SDG 6.6.1 "Change in extent of water-related ecosystems over time" NL, for example, Estimation of maximum floodplain extent estimated using satellite imagery, Inner Niger Delta (Mali). |
| Benchmark (standard)—Quantitative value given to the indicator that is used to assess performance to reach the target | 2030 Agenda has not provided benchmarks other than qualitative changes, for example, SDG 6.2 " … halving the proportion of untreated wastewater … … …." GL, for example, the IPCC global temperature target of <2 °C above pre-industrial levels. NL, for example, Maximum floodplain extent of the Inner Niger Delta is maintained at >85% of natural average. |

The hierarchy starts with the most over-arching term (vision) and ends with the most directed and specific (benchmark). GL = Global level; NL = National level. National level use will be unique for each country and will depend on the national context.

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
