# Peer review of "Defining and Quantifying National-Level Targets, Indicators and Benchmarks for Management of Natural Resources to Achieve the Sustainable Development Goals"

_sustainability, doi:10.3390/su11020462_

Round 1

Reviewer 1 Report

Generally the paper is well written and clearly constructed. The focus on national implementation of SDGs is current and important.

Title & Abstract

- it is given the impression of something generic but the methodology is described to apply for two basins in South-Africa, for natural resource-security focussed SDGs => this restriction should be made clear from the beginning

- the focus is on the country perspective but the main authors are representing global actors, a top-down writing approach, stakeholders are mentioned as important partners but they have not been actively involved in the writing process => how does this affect the credibility of the methodology proposed (that is the backbone of the paper) when it comes to implementation in practice?

Introduction

- the introduction gives important information on the process on SDG implementation both on the global and the national levels

- it is good to be specific on the terminology and the definition of sustainability

- interesting references and links are included

Material and methods

- chapter 2 reads a bit like a recipe, maybe a graphic could have been developed to illustrate the different steps and to give an overview?

- the aim "... assist national governments to quantitatively describe natural resources in order to benchmark " is very generic => which countries? all countries? also those who already have an assessment procedure in place? for what purpose exactly? => here some more precision is required

- what kind of mandate does the desribed procedure have or need? or is it just meant to give ideas? => also here be more precise

Results

- the example of South-Africa visualizes the procedure but the current description also raises questions that are not yet addressed (applies also for the Discussion)

- mainly it is a description of a classification system but does not show anything on prioritisation consequences for the subbasins concerning e.g. monitoring, also the part on how the recommendations get to the government gazette that enters the benchmarks into law are not described in detail => yet these would be the novelties when putting the methodology into practice

Discussion

- the impression is given that the gain so far of this methodology for South Africa is the quantification and publication of benchmarks for the selected basins, is this correct?

- for countries with existing assessment systems this would not be a gain, what does it mean for these countries?, can they learn something from the approch described here? => mentioning that there are other systems (lines 397-442) is too little as no link between the methods is made => e.g. for the national voluntary reporting, do the authors expect that this methodology brings some benefits or would the countries applying e.g. WFD just ignore it and continue using what they already have? => would a merger make sense? or an iteraction on some forum (which?) to exchange information on the status of national reporting?

Conclusions

- I would expect not only to state that guidance is provided but to be clear about the relation to already existing national assessment systems and the legitimacy/authorization needed to use this kind of approach => be more explicit concerning countries with and without an existing assessment systems

Author Response

Reviewer 1

Generally the paper is well written and clearly constructed. The focus on national implementation of SDGs is current and important.

Title & Abstract

- it is given the impression of something generic but the methodology is described to apply for two basins in South-Africa, for natural resource-security focussed SDGs => this restriction should be made clear from the beginning

Response – the intention is to provide a generic methodology.  The South African example was a predecessor that only guided the development of the proposed method–thusit cannot be said that the South African example was a case study for the method. 

              Added textLine 40:The procedure builds on precedent set in other countries, but in particular on a procedure developed for the setting of Resource Quality Objectives in South Africa (Dickens et al, 2011).

- the focus is on the country perspective but the main authors are representing global actors, a top-down writing approach, stakeholders are mentioned as important partners but they have not been actively involved in the writing process => how does this affect the credibility of the methodology proposed (that is the backbone of the paper) when it comes to implementation in practice?

Response – the main authors do represent global actors but have worked and indeed developed the previous method (Dickens et al 2011) while working on contract to a national government and based within that country.  The approach has also been used in Lesotho and Mali and also is recommended by the Nile Basin Initiative for all riparian countries in the Nile.  During the conceptualisation of  the previous method (see response comment above)right back from 2009 until the present,there was extensive stakeholder consultation which is documented in References 20 and 21 in the draft paper, and also during the SDG development process–where stakeholders included various members of the UN working on Goal 6 (with GEMI) and also in 10 countries where workshops were held with the country governments to review indicators for SDG 6.6.1 and 6.3.2.  While not explicitly stakeholders for setting targets, in GEMI and in each country the issue was indicators for monitoring of sustainability, and thus targets were a point of discussion. So while it is agreed thatt he paper has been written in a top-down way,this follows recognition of the needs of many countries following extensive consultation although this was not directly linked.  It is suggested that there is sufficient precedent to provide the required credibility. 

Introduction

- the introduction gives important information on the process on SDG implementation both on the global and the national levels

- it is good to be specific on the terminology and the definition of sustainability

- interesting references and links are included

Material and methods

- chapter 2 reads a bit like a recipe, maybe a graphic could have been developed to illustrate the different steps and to give an overview?

Response:  A figure has been inserted at Line 236.

- the aim "... assist national governments to quantitatively describe natural resources in order to benchmark " is very generic => which countries? all countries? also those who already have an assessment procedure in place? for what purpose exactly? => here some more precision is required

Response:  Yes all countries who need assistance.  Very few countries already have a procedure in placeto set numeric targets for natural resources.  Even those that do, as mentioned in the paper e.g. South Africa, Australia and New Zealand, the Water Framework Direcive, do not have procedures that are directly linked to the SDGs.

- what kind of mandate does the desribed procedure have or need? or is it just meant to give ideas? => also here be more precise

Response:  This is just a scientific paper that any country is free to make use of or not.  There is no mandate from the UN or anyone else.

               Added text:Line 231:This procedure is suitable for any country wishing to set target and benchmarks for the management of natural resources, and is of particular value in aligning national targets with Agenda 2030 on Sustainable Development. 

Results

- the example of South-Africa visualizes the procedure but the current description also raises questions that are not yet addressed (applies also for the Discussion)

- mainly it is a description of a classification system but does not show anything on prioritisation consequences for the subbasins concerning e.g. monitoring, also the part on how the recommendations get to the government gazette that enters the benchmarks into law are not described in detail => yet these would be the novelties when putting the methodology into practice

Response:  The South African example is a precedent but not directly – i.e. the new method proposed in this paper builds on that prior example. Because of space limitations, it was decided not to describe the South African example in great detail.

It would be possible to insert a map illustrating prioritisation of sub-basins and resource units to be used for monitoring – however space precludes this. The source documents can be referred to.  If a country were to implement this method–itwould be useful for them to examine the source documents of the South African example. The same applies to the gazetting process which is beyond this paper.

Discussion

- the impression is given that the gain so far of this methodology for South Africa is the quantification and publication of benchmarks for the selected basins, is this correct?

Response:  The RQO method in South Africa,which was used as precedent for this method, does quantify and publish benchmarks for basins–that is true.  But the method in this paper is substantially advanced in that it is for global use and for all resources, to fit with the SDG Agenda.

- for countries with existing assessment systems this would not be a gain, what does it mean for these countries?,

Response:  Very few countries have a procedure for setting benchmarks for natural resources.  If indeed they have, then they would be interested in this approach and could test their procedure against it. can they learn something from the approah described here?

=> mentioning that there are other systems (lines 397-442) is too little as no link between the methods is made => e.g. for the national voluntary reporting, do the authors expect that this methodology brings some benefits or would the countries applying e.g. WFD just ignore it and continue using what they already have? => would a merger make sense? or an iteraction on some forum (which?) to exchange information on the status of national reporting?

Response:  While a merger of the method would be a win for the publication, this is just a scientific paper and has no mandate to get any country to change its approach or procedures.  As noted above, there are very few countries that have similar procedures in place, so this proposed method should be of great value to them.

Conclusions

- I would expect not only to state that guidance is provided but to be clear about the relation to already existing national assessment systems and the legitimacy/authorization needed to use this kind of approach => be more explicit concerning countries with and without an existing assessment systems

Added text: line 398: Most countries do not however have such procedures and thus may find the procedure in this paper useful.  Where a country does have a procedure, this paper may be of value in reviewing their approach to confirm that it addresses the kind of objectives management approach that is necessary for Agenda 2030 to be a success.

Reviewer 2 Report

The article proposes a methodology to develope targets and indicators for SDGs that are based on the protection and sustainable use of natural resources, and that require monitoring the state of natural resources. However, it is assured that it can be transferred to all SDGs.

In the conclusions and generally throughout the text, the authors argue that the 2030 agenda leaves out or without guidance the SDGs focused on resources, even indicating that there is no guide or procedure available to do this. However, the UN has always established guidelines and working standards (even when it was responsible to do so with the MDGs), through its various international institutions, working commissions, agencies, etc. For example, there is a guide for the integrated monitoring of Sustainable Development Goal 6 on water and sanitation and the same occurs to the rest of the SDGs.

Due to all of the above, the authors should review the text taking into account the methodologies or guidelines for SDGs and/or indicators that already exist. For this purpose, some of the following proposals would be sufficient:

1. That the authors would describe (as they already do) their experiences in the Vaal and Olifants rivers. These experiences were before Agenda 2030 came into force and they could compare the procedure they followed at that time, with the monitoring guidelines currently proposed by the United Nations, and thus, propose a discussion.

2. That the authors simply describe their experiences and results, encouraging them to continue with similar works under the new indications or guidelines of the SDGs.

The article is fine in general and has a good structure. However, the issue I referred to above needs to be resolved because the text cannot be published by ensuring that there are no SDGs guidelines focused on natural resources.

Another less important matter is that on occasion, the authors could have cited some document without referencing it in the bibliography, for example, when they refer to the U.S. Environmental Protection Agency. In any case, it is not an important issue.

Yours Respectfully

Author Response

The efforts by the reviewer are greatly appreciated.  Here is the response.

Reviewer 2

Comments and Suggestions for Authors

The article proposes a methodology to develope targets and indicators for SDGs that are based on the protection and sustainable use of natural resources, and that require monitoring the state of natural resources. However, it is assured that it can be transferred to all SDGs.

Response:  As indicated in line 42 and elsewhere, this procedure is focussed on natural resources. Line 186 notes that “Those SDG targets that are most suited to implementation of this procedure are those that seek to protect land and water resources (see below) while other indicators may also benefit from the essence of this approach” which is the only suggestion that it could be transferred to all SDGs.  Considerable adaptation would be neededfor thatto be done.

In the conclusions and generally throughout the text, the authors argue that the 2030 agenda leaves out or without guidance the SDGs focused on resources, even indicating that there is no guide or procedure available to do this. However, the UN has always established guidelines and working standards (even when it was responsible to do so with the MDGs), through its various international institutions, working commissions, agencies, etc. For example, there is a guide for the integrated monitoring of Sustainable Development Goal 6 on water and sanitation and the same occurs to the rest of the SDGs.

Response:  Yes the UN has published numerous guidelines, but none of these address the issue of countries setting quantitative benchmarks at anationallevel.  The IntegratedMonitoringGuideline for SDG6 is a good example, it is very generic and does not provide any of the guidance provided in this paper.

Due to all of the above, the authors should review the text taking into account the methodologies or guidelines for SDGs and/or indicators that already exist. For this purpose, some of the following proposals would be sufficient:

1. That the authors would describe (as they already do) their experiences in the Vaal and Olifants rivers. These experiences were before Agenda 2030 came into force and they could compare the procedure they followed at that time, with the monitoring guidelines currently proposed by the United Nations, and thus, propose a discussion.

Response: Further to theresponse above, there is little value in reviewing the UN guidelines as these do not provide sufficient detail. 

2. That the authors simply describe their experiences and results, encouraging them to continue with similar works under the new indications or guidelines of the SDGs.

Response:  The South African example is not exactly the same method but is a predecessor–thusthere would be limited value in describing in great detail.  It is recommended that where a country wishes to implement the method presented here, that they could review the South African reports just to get some detailed ideas on how to do this. 

Added text:  Line 397:.  These country examples provide useful guidance to a country implementing the approach presented in this paper,as their examples provide detail of many of the recommended steps.  Most countries do not however have such procedures and thus may find the procedure in this paper useful. 

The article is fine in general and has a good structure. However, the issue I referred to above needs to be resolved because the text cannot be published by ensuring that there are no SDGs guidelines focused on natural resources.

Response:  As noted above, there are no UN guidelines that go into the detail that is required to set numerical benchmarks.

Another less important matter is that on occasion, the authors could have cited some document without referencing it in the bibliography, for example, when they refer to the U.S. Environmental Protection Agency. In any case, it is not an important issue.

Response:  This is reference no. 28.  The reference has been clarified in Line 451.

Round 2

Reviewer 2 Report

The corrections are sufficient. 

The title of the manuscript is too generic, it could refer more specifically to the work done. However, its modification is not necessary.